# PET Imaging of the Neurotensin Targeting Peptide NOTA-NT-20.3 Using Cobalt-55, Copper-64 and Gallium-68

**DOI:** 10.3390/pharmaceutics14122724

**Published:** 2022-12-06

**Authors:** Hailey A. Houson, Volkan Tekin, Wilson Lin, Eduardo Aluicio-Sarduy, Jonathan W. Engle, Suzanne E. Lapi

**Affiliations:** 1Department of Radiology, University of Alabama at Birmingham, Birmingham, AL 35294, USA; 2Department of Medical Physics, University of Wisconsin, 1111 Highland Avenue, Madison, WI 53705, USA; 3Department of Radiology, University of Wisconsin, 1111 Highland Avenue, Madison, WI 53705, USA

**Keywords:** neurotensin, PET, cobalt-55, copper-64, gallium-68

## Abstract

Introduction: Neurotensin receptor 1 (NTSR1) is an emerging target for imaging and therapy of many types of cancer. Nuclear imaging of NTSR1 allows for noninvasive assessment of the receptor levels of NTSR1 on the primary tumor, as well as potential metastases. This work focuses on a the neurotensin peptide analogue NT-20.3 conjugated to the chelator NOTA for radiolabeling for use in noninvasive positron emission tomography (PET). NOTA-NT-20.3 was radiolabeled with gallium-68, copper-64, and cobalt-55 to determine the effect that modification of the radiometal has on imaging and potential therapeutic properties of NOTA-NT-20.3. Methods: In vitro assays investigating cell uptake and subcellular localization of the radiolabeled peptides were performed using human colorectal adenocarcinoma HT29 cells. In vivo PET/CT imaging was used to determine the distribution and clearance of the peptide in mice bearing NTSR1 expressing HT29 tumors. Results: Cell uptake studies showed that the highest uptake was obtained with [^55^Co] Co-NOTA-NT-20.3 (18.70 ± 1.30%ID/mg), followed by [^64^Cu] Cu-NOTA-NT-20.3 (15.46 ± 0.91%ID/mg), and lastly [^68^Ga] Ga-NOTA-NT-20.3 (10.94 ± 0.46%ID/mg) (*p* < 0.001). Subcellular distribution was similar across the three constructs, with the membranous fraction containing the highest amount of radioactivity. In vivo PET/CT imaging of the three constructs revealed similar distribution and tumor uptake at the 1 h imaging timepoint. Tumor uptake was receptor-specific and blockable by co-injection of non-radiolabeled NOTA-NT-20.3. SUV ratios of tumor to heart at the 24 h imaging timepoint show that [^55^Co] Co-NOTA-NT-20.3 (20.28 ± 3.04) outperformed [^64^Cu] Cu-NOTA-NT-20.3 (6.52 ± 1.97). In conclusion, our studies show that enhanced cell uptake and increasing tumor to blood ratios over time displayed the superiority of [^55^Co] Co-NOTA-NT-20.3 over [^68^Ga] Ga-NOTA-NT-20.3 and [^64^Cu] Cu-NOTA-NT-20.3 for the targeting of NTSR1.

## 1. Introduction

The neurotensin peptide is a 13-amino-acid peptide which binds to neurotensin receptors 1–3 (NTSR1–3) [1]. Neurotensin receptors 1–3 are involved in the regulation of multiple biologic functions and are most highly expressed in the gut and central nervous system [2,3]. Of the subtypes, neurotensin receptor 1 (NTSR1) is preferentially upregulated in cancers and is involved in cancer development and progression [4]. A significant variety of cancers have been found to overexpress NTSR1, including pancreatic, colorectal, and breast cancers, among others [5,6,7]. 

Neurotensin receptor imaging has emerged as a promising target for imaging and therapy. The endogenous neurotensin peptide has a short half-life in blood (<6 min), limiting its direct radiolabeling and use [8,9]. Most developments in neurotensin receptor imaging have been focused on increasing serum stability and improving specificity for NTSR1 [1]. Previously, neurotensin receptor binding peptides have been investigated as imaging and therapeutic agents when radiolabeled with technetium-99m, indium-111, gallium-68, copper-64, actinium-225, and leutetium-177 [10,11,12,13]. Specifically, significant work by the Gruaz-Guyon group has resulted in the development of the neurotensin analog NT-20.3, which shows excellent characteristics including high serum stability (>250 h), a low IC50 (2.2 nM), and high rates of internalization [14,15].

A radionuclide theranostic pair involves using two different radioisotopes with one enabling pretreatment imaging and the second acting as a radiotherapy [16]. Interest in radionuclide theranostic pairs has been fueled by the success of the [^68^Ga] Ga-DOTATATE/[^177^Lu] Lu-DOTATATE pair [17,18]. While the gallium-68 (t_1/2_ = 68 min)/leutetium-177 (t_1/2_ = 6.7 d) pair has been successful for DOTATATE, it is unclear if this relationship is broadly applicable for other tracers [19]. However, the in vivo characteristics of a matched radionuclide pair where both the imaging and therapeutic isotopes are of the same element allow for the dosimetry calculated in imaging to exactly match that of the therapeutic radioisotope [16]. 

Matched theranostic pairs such as copper-64 (t_1/2_ = 12.7 h)/copper-67 (t_1/2_ = 2.58 d) allow imaging and therapy to occur with an identical molecule. Additionally, there is interest in the theranostic pair of cobalt-55 (t_1/2_ = 17.5 h)/cobalt-58m (t_1/2_ = 9.1 h). Cobalt-55 is a positron emitter which has been studied most as [^55^Co] Co-DOTATATE [20]. Promising in vitro results have also been observed with the Auger–Meitner emitter [^58m^Co] Co-DOTATATE [21].

Many groups have shown that changing the radiometal can have a significant impact on binding efficacy and off-target uptake in radiometal peptide imaging [22,23]. Alterations in the chelator and metal incorporated into NT-20.3 peptide have been observed to change its binding to NTSR1 [15]. In this study, we investigated the in vitro and in vivo characteristics of NOTA-NT-20.3 radiolabeled with three positron-emitting radiometals: gallium-68, cobalt-55, and copper-64.

## 2. Methods

### 2.1. Materials 

All reagents and buffers purchased were of trace metal grade and prepared in Milli-Q water. 

NOTA conjugated NT-20.3 peptide (Ac-Lys (NOTA)-Pro-NMeArg-Arg-Pro-Tyr-Tle-Leu) was produced by CPC Scientific (San Jose, CA, USA). 

Isotope Production: Cobalt-55 and copper-64 were produced at the University of Alabama at Birmingham cyclotron facility on an ACSI TR24 cyclotron, or at the University of Wisconsin, Madison cyclotron facility on a GE PETtrace. 

Cobalt-55 was produced via the ^58^Ni (p,α)^55^Co or the ^54^Fe(d,n)^55^Co reactions, as has been previously published [24,25]. For the former, nickel-58 (Isoflex, San Francisco, CA, USA) was electroplated onto a 1 mm gold disc and placed behind a 0.75 mm aluminum degrader. The target was bombarded with an 18 MeV proton beam (13.4 MeV on the target) at 40 uA for 4 h. After bombardment, the coin was placed in a beaker, and the nickel-58 and cobalt-55 were dissolved in 9 M HCl. The dissolved target was loaded onto a column composed of 2.5 g of AG1x8 resin (Bio-Rad laboratories, Hercules, CA, USA), and target material was eluted with 50 mL of 9 M HCl. Cobalt-55 was eluted with 2–2 mL aliquots of 0.5 M HCl. The second aliquot was combined with 9 M HCl and loaded onto a second column containing 2.5 g of AG1x8. The washing and eluting procedure was repeated. The second elution fraction was dried at 200 °C and reconstituted in 100 μL of 0.1 M HCl. For the latter production route, iron-54 was electroplated onto silver backings and irradiated with 8 MeV deuterons and currents up to 60 µA. The iron-54 was dissolved in 6 M HCl and oxidized to Fe (3+) with H_2_O_2_. This solution was diluted to 95% EtOH/0.3 M HCl and loaded onto AG1x8 resin. Cobalt was subsequently eluted in 4 M HCl, concentrated to 8 M HCl, and loaded onto 300 mg DGA extraction resin. Purified cobalt-55 was then eluted in 2 M HCl, dried, and reconstituted in 0.1 M HCl for labeling. 

Copper-64 was produced via the ^64^Ni (p,n)^64^Cu reaction [26]. Nickel-64 (Isoflex, San Francisco, CA, USA) was electroplated into a 1 mm thick gold backing and placed behind a 1 mm aluminum degrader. The target was bombarded at 18 MeV on the degrader (13 MeV on target) and 40 uA for 2–4 h. After bombardment, the target was dissolved in 9 M HCl, loaded on a 2.5 g AG1x8 resin column, and washed with 15 mL of 9 M HCl. Co-produced cobalt-61 was eluted with 4 mL of 4 M HCl. Copper-64 was eluted with 5 mL of 0.1 M HCl, dried at 100 °C under vacuum, and reconstituted in 0.1 M HCl.

Gallium-68 was obtained from a germanium-68/gallium-68 generator (Eckert and Ziegler, Valencia, CA, USA). To elute gallium-68, 10 mL of 0.1 M HCl was passed through the generator and gallium-68 was trapped on a 30 mg Strata-X-C cartridge (Phenomenex, Torrance, CA, USA). Gallium was eluted from the cartridge using 400 μL of acetone. 

### 2.2. Radiolabeling of NOTA-NT-20.3

Radiolabeling of NOTA-NT-20.3 with cobalt-55 and copper-64 was performed using two methods. In the first, the tracer (37 MBq of [^55^Co] CoCl_2_ or 74 MBq of [^64^Cu] CuCl_2_) was reacted with 14.4 nmol NOTA-NT-20.3 (20 μg) and 50 μL of NaOAc buffer, pH 3.5. The reaction mixture was heated at 98 °C for 30 min and shaken at 700 RPM. QC was performed on an Agilent HPLC using a Phenomenex Kinetex C18 column and a gradient of 5–50% ACN over 30 min. The in vitro and in vivo data reported below use [^55^Co] Co-NOTA-NTS-20.3 radiolabeled using this method. 

A high specific activity method was also developed. [^55^Co] Co-NOTA-NT-20.3 was radiolabeled at 7.4 MBq/nmol of ligand in pH 4.5 NaOAc buffer and heated at 95 °C for 60 min with 2 mg/mL gentisic acid to inhibit radiolysis. Radiochemical purity was assessed by radio-HPLC using a reverse-phase 250 × 4.60 mm C18 5μm 100Å column (DIONEX), and the gradient is outlined in Figure 1. The HPLC data in Appendix A show [^55^Co] Co-NOTA-NT-20.3 radiolabeled using this method. 

Radiolabeling of NOTA-NT-20.3 with gallium-68 was performed with 14.4 nmol (20 μg), 74 MBq of GaCl_3_, and 50 μL of NaOAc buffer, pH 3.5. The reaction mixture was heated at 98 °C for 10 min and shaken at 700 RPM. QC was performed using iTLC paper and developed in a 1:1 mixture of methanol and 10% *w*/*v* ammonium acetate. 

### 2.3. In Vitro Studies

Serum stability of NOTA-NT-20.3 radiolabeled with gallium-68, copper-64, and cobalt-55 was assessed by incubation of the radiotracer with human serum (Thermo Fisher, Waltham, MA, USA) or PBS at 37 °C. At the indicated timepoints, 20 μL of serum was removed and combined with 20 μL of methanol to precipitate proteins. Supernatant was injected into HPLC to determine the intact percent of the peptide.

Human colorectal adenocarcinoma cells HT29 (ATCC, Manassas, VA, USA) were grown in McCoy’s 5A medium supplemented with 10% FBS and gentamycin. Human colorectal adenocarcinoma line Caco-2 (ATCC, Manassas, VA, USA) was cultured in EMEM supplemented with 10% FBS and gentamicin.

For cell uptake assays, 250,000 cells were plated in 12 well plates 24–48 h before the start of the assay. Radiolabeled NOTA-NT-20.3 was incubated at 10 nM in 400 μL of media per well (n = 6/group). In blocking wells, cold NOTA-NT-20.3 was added to a concentration of 1 μM. Cells were incubated at 37 °C for 1 h. Cells were washed with 1 mL ice cold PBS twice, then lysed with 200 μL of 0.2 M NaOH. After gamma counting, BCA was used to quantify the protein in each well.

For subcellular fractionation assays, HT-29 cells were grown to confluence on T-175 flasks. To each flask, 15 mL of 100 nM radiolabeled NOTA-NT-20.3 was added and incubated at 37 °C for 1 h. After 1 h, the flask and cells were washed twice with cold PBS and cells were detached with trypsin. Cells were collected in a 50 mL centrifuge tube with PBS and centrifuged at 500 RPM for 5 min. Trypsin and PBS were removed, and cells were washed once more with cold PBS. PBS was removed, and 3–100 μL aliquots were taken from the cell pellet. Cells were processed according to the Thermo Fisher subcellular fractionation kit for cultured cells (MA). Resulting fractions were quantified for radioactivity using a gamma counter. 

### 2.4. In Vivo Experiments

All animal procedures were approved by the University of Alabama at Birmingham IACUC, animal protocol number IACUC-22618. Male athymic nude mice (Charles River, Wilmington, MA, USA) were allowed to acclimate for at least 72 h before any procedures were performed. For tumor implantation, mice were subcutaneously injected with 300,000 HT29 cells 3 weeks before imaging. For imaging, mice were injected with 1 μg of radiolabeled NOTA-NT-20.3 or 1 μg of radiolabeled NOTA-NT-20.3 and 200 μg non-radiolabeled NOTA-NT-20.3 (n = 4 per group) and imaged at 1 h (gallium-68, copper-64, cobalt-55), 4 h, and 24 h (copper-64 and cobalt-55). Mice were imaged on a GNEXT microPET/CT (Sophie, Springfield, VA, USA). After the last imaging timepoint, mice were euthanized and organs were taken for biodistribution. Organ weights and radioactive counting were collected on a Hidex gamma counter (Lablogic, Clair-Mel City, FL, USA). Images were processed and SUVs were calculated using VivoQuant software (Invicro, Boston, MA, USA).

### 2.5. Statistical Analysis

Comparison of cell uptake in HT29 cells was performed with an ordinary one-way ANOVA and Tukey’s multiple comparison test. Comparison of the SUV ratio and BioD data was performed with an ordinary two-way ANOVA with Tukey’s multiple comparison test with pooled variance. Statistical significance is indicated as *p* < 0.05 = *, *p* < 0.01 = **, *p* < 0.001 = ***, and *p* < 0.0001 = ****.

## 3. Results/Discussion

Copper-64 and cobalt-55 were produced by the UAB cyclotron facility with a typical specific activity of 37 GBq/μmol and 1.1 GBq/μmol, respectively. Copper-64 and cobalt-55 were used on the day after production, enabled by their relatively long half-life, whereas gallium-68 was used immediately after elution (Table 1).

Radiolabeling routinely yielded [^68^Ga] Ga-NOTA-NT-20.3, [^64^Cu] Cu-NOTA-NT-20.3, and [^55^Co] Co-NOTA-NT-20.3, which was above 95% radiochemical purity within 30 min. We achieved a molar activity of 5.13 MBq/nmol for [^68^Ga] Ga-NOTA-NT-20.3 and [^64^Cu] Cu-NOTA-NT-20.3 and a molar activity of 2.56 MBq/nmol for [^55^Co] Co-NOTA-NT-20.3 (Appendix A). When this material was assessed for stability in PBS and human serum, we observed that all of the radiolabeled peptides were stable out to the duration assessed in the imaging studies (Table 2). These results corroborate that the modifications included in NT-20.3 enhanced serum stability as compared to endogenous neurotensin peptide [14].

Cell uptake studies were performed in NTSR1 expressing HT29 cells and nonexpressing Caco-2 cells at 37 °C and 4 °C (Figure 1). HT29 cells that were incubated with unlabeled blocking peptide showed significant decreases in uptake as compared to their non-blocked counterpart. Using [^68^Ga] Ga-NOTA-NT-20.3, we observed uptake at 1 h of 10.9 ± 0.5%/mg, compared to 15.5 ± 0.9%/mg with [^64^Cu] Cu-NOTA-NT-20.3 and 18.7 ± 1.3%/mg with [^55^Co] Co-NOTA-NT-20.3. The difference between each of the groups was statistically significant at a *p* value of <0.0001 (Appendix A). Additionally, incubation at 37 °C showed a significant increase in cell-associated radioactivity as compared to cells incubated at 4 °C. This is likely due to the rapid internalization of neurotensin at 37 °C and the lack of internalization at 4 °C [14]. Expectedly, Caco-2 cells showed low levels of uptake in all conditions tested. It is currently unclear why NOTA-NT-20.3 labeled with cobalt-55 illustrates increased uptake over the peptide labeled with copper-64 or gallium-68. It is not aligned with the trend of atomic radius (Cu < Ga < Co) or with the atomic charge state (Cu^2+^, Co^2+/3+^, Ga^3+^). 

In the subcellular fractionation assay, we consistently observed the highest activity in the membrane-bound fraction, which includes plasma membrane and membranes of membrane-bound organelles in the cytoplasm (Figure 2). The highest uptake was observed with [^55^Co] Co-NOTA-NT-20.3 (76.44 ± 2.57%), followed by [^64^Cu] Cu-NOTA-NT-20.3 (71.51 ± 0.68%) and [^68^Ga] Ga-NOTA-NT-20.3 (66.17 ± 7.03%). The nuclear-bound and cytoplasmic fractions were observed to contain the second highest associated activity. These findings agree with results observed by Boudin et al., who showed that neurotensin binding peptide was localized primarily in the cytoplasmic membrane and early endosomes [27]. 

The subcellular localization of the radiopeptide in the cell may influence its therapeutic translation. Auger–Meitner emitters deposit their energy in a short range and need to come in close proximity to DNA in order to cause strand breaks [28,29]. The use of NOTA-NT-20.3 as a therapeutic with an Auger–Meitner emitter such as cobalt-58m may be reliant on significant translocation of [^58m^Co] Co-NOTA-NT-20.3 to the nucleus. We observed approximately 10% of the cell-associated radioactivity in the nuclear soluble fraction for each agent within 1 h of incubation. Previously, nuclear translocation of [^57^Co] Co-DOTATATE in AR42J cells was observed to constitute 2–4% of cell associated activity [21]. Effective cell killing was observed using [^58m^Co] Co-DOTATATE in the same model [21]. With approximately 10% nuclear translocation observed using NOTA-NT-20.3, we are optimistic that it may serve as an effective therapeutic when radiolabeled with cobalt-58m. 

PET/CT imaging of the radiolabeled NOTA-NT-20.3 peptide shows significant tumor accumulation within 1 h post-injection in the 1 μg injection groups (Figure 3). In the 200 μg injection groups, we observed very low tumor uptake, presumably as available binding sites for NTSR1 were blocked. 

At the 4 h PI timepoint, images show sustained tumor uptake and additional washout of the radiolabeled peptide from non-target organs (Figure 4). At this timepoint, differences between imaging agents were apparent. In the mice injected with [^64^Cu] Cu-NOTA-NT-20.3, liver uptake was observed in the 1 μg injection group. Most likely, this was the result of the release of free copper-64 from the imaging agent, which was known to be sequestered by the liver [30]. Interestingly, this was observed less in the blocking group, as both liver and kidney accumulation were reduced as compared to the low dose group. Imaging with [^55^Co] Co-NOTA-NT-20.3 showed significant tumor uptake in the 1 μg injection group, which was not observed in the high-mass (blocking) group. Similarly to [^64^Cu] Cu-NOTA-NT-20.3, there was reduced kidney uptake in the 200 μg injection group.

Tumor accumulation of both [^64^Cu] Cu-NOTA-NT-20.3 and [^55^Co] Co-NOTA-NTS-20.3 was sustained at the 24 h PI timepoint in the 1 μg injection group (Figure 5). Images from mice injected with [^64^Cu] Cu-NOTA-NT-20.3 showed clear liver accumulation in both the 1 μg and 200 μg injection groups, which is attributable to decomplexation of copper from the ligand and subsequent uptake in the liver. In contrast, cobalt imaging did not show increased uptake in any nontarget organs at the 24 h timepoint. Similar to observations at earlier timepoints, reduced kidney uptake was observed in the 200 μg mass dose group. It is unclear if the reduction in kidney uptake seen in the 200 μg mass dose group is due to the blocking of binding sites in clearance organs. NTSR1 is expressed primarily in the brain and intestines, so the reduction in kidney uptake was unexpected [31]. It may be possible that there is a mass dose which yields sustained tumor uptake while blocking the accumulation in non-target organs.

SUV analysis of selected organs over the 1 h, 4 h, and 24 h imaging timepoints shows the clearance of the radiolabeled NOTA-NT-20.3 from the blood as observed in the heart SUV (Figure 6). [^55^Co] Co-NOTA-NT-20.3 images illustrate the blood clearance in the heart SUV through the 1 h (0.033 ± 0.002), 4 h (0.015 ± 0.002), and 24 h (0.004 ± 0.001) timepoints. Liver uptake was observed to be stable in the [^64^Cu] Cu-NOTA-NT-20.3 imaging group through the 1 h (0.123 ± 0.029), 4 h (0.178 ± 0.042), and 24 h (0.168 ± 0.056) imaging timepoints. 

At the 1 h imaging timepoint, [^68^Ga] Ga-NOTA-NT-20.3 was observed to have the highest tumor SUV (0.392 ± 0.083), followed by [^64^Cu] Cu-NOTA-NT-20.3 (0.315 ± 0.081) and [^55^Co] Co-NOTA-NT-20.3 (0.257 ± 0.055) (Figure 6A). Co-injection of 1 μg of the radiolabeled NOTA-NT-20.3 with a 200 μg non-labeled dose resulted in apparent delayed blood clearance of the peptide. This was most apparent in the [^55^Co] Co-NOTA-NT-20.3 images at 1 h PI, where heart SUV in the 1 μg group was 0.033 ± 0.002, which increased in the 200 μg group to 0.230 ± 0.068 (Figure 6A). By 4 h PI, the heart SUV in the 1 μg (0.015 ± 0.002) and 200 μg (0.017 ± 0.008) groups for mice imaged with [^55^Co] Co-NOTA-NT-20.3 were close in value (Figure 6B). It is possible that the large mass dose of NOTA-NT-20.3 displayed a pharmacologic action. Other groups have observed apparent delayed clearance with the addition of a blocking dose of NOTA-NT-20.3 [12,32]. 

Both tumor to heart and tumor to muscle ratios continued to increase for [^55^Co] Co-NOTA-NT-20.3 throughout selected imaging timepoints (Figure 6D,E). In contrast, the tumor to muscle ratio for [^64^Cu] Cu-NOTA-NT-20.3 increased while the tumor to heart ratio was static for the duration studied. This may be due to the degradation of the imaging agent in vivo. Free copper and free cobalt are known to be taken up in large quantities by the liver [30,33]. SUV data at 24 h and SUV ratios appear to suggest that cobalt may be decomplexing to a lesser degree than copper, which is in agreement with other reports [33]. 

Biodistribution was assessed in select organs after the final imaging timepoint (Figure 7). We observed that kidney retention of the imaging agents was much higher in the [^55^Co] Co-NOTA-NT-20.3 mice than in [^64^Cu] Cu-NOTA-NT-20.3 at the 24 h timepoint. Kidney uptake could be indicative of the clearance of free cobalt, as has been observed before, or could be attributed to the presence of a Co^3+^ complex [33]. 

The %ID/g values for the tumor decreased between the 1 h biodistribution timepoint ([^68^Ga] Ga-NOTA-NT-20.3) and the 24 h timepoint (^55^Co- and ^64^Cu-labeled NOTA-NT-20.3), but during that time, significant clearance from the blood was also observed. This may mean that continued monitoring of the mice in the ^55^Co- and ^64^Cu-labeled NOTA-NT-20.3 groups may result in higher tumor to blood ratios and better images longer than 24 h post-injection. 

## 4. Conclusions

The neurotensin receptor has emerged as an exciting target for nuclear imaging and radiotherapy, and has been investigated using a variety of chelators and radiometals. We observed that changing the radiometal in an otherwise unchanged small molecule had multiple impacts on its function as an imaging agent. In vitro, we observed that [^68^Ga] Ga-NOTA-NT-20.3 displayed the lowest total uptake in the cells, and the half-life of gallium-68 was too short to perform imaging at protracted timepoints. Of the three radionuclides studied, gallium-68 has the highest positron emission energy (829 keV), but this did not seem to have a negative impact on image quality in our PET/CT images. The comparatively longer half-lives of copper-64 (12.7 h) and cobalt-55 (17.5 h) allowed for imaging to occur out to 24 h post-injection. It appears that [^55^Co] Co-NOTA-NT-20.3 may be superior for imaging neurotensin as compared to [^64^Cu] Cu-NOTA-NT-20.3. In cell studies, [^55^Co] Co-NOTA-NT-20.3 uptake was superior to [^64^Cu] Cu-NOTA-NT-20.3. In PET images, 24 h SUV uptake in the tumors was similar between agents, but tumor to heart ratios continued to increase. As the tracer was metabolized and the radiometal was released, copper was observed to be taken up by the liver while cobalt was not. In this study, [^55^Co] Co-NOTA-NT-20.3 outperformed [^64^Cu] Cu-NOTA-NT-20.3 and [^68^Ga] Ga-NOTA-NT-20.3 in both cell uptake studies and PET imaging studies. 

## Figures and Tables

**Figure 1 pharmaceutics-14-02724-f001:**
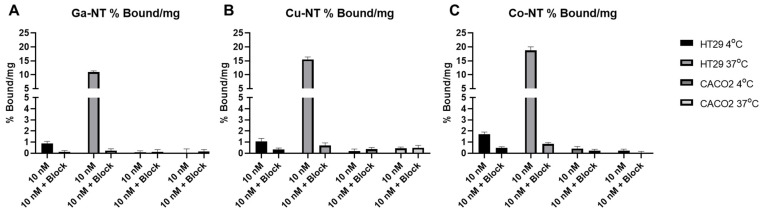
Cell uptake of NOTA-NT-20.3 labeled with gallium-68, copper-64, and cobalt-55 shows specific uptake in neurotensin expressing HT29 cells and minimal uptake in low neurotensin expressing Caco2 cells. Cell uptake was investigated using [^68^Ga] Ga-NOTA-NT-20.3 (**A**), [^64^Cu] Cu-NOTA-NT-20.3 (**B**), and [^55^Co] Co-NOTA-NT-20.3 (**C**).

**Figure 2 pharmaceutics-14-02724-f002:**
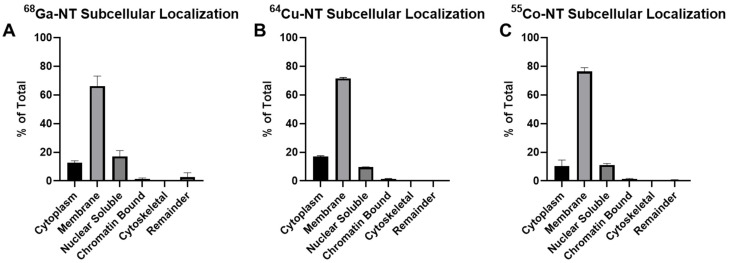
Subcellular localization of NOTA-NT-20.3 labeled with gallium-68, copper-64, and cobalt-55 shows primarily membrane localization after 1 h of incubation. Cell localization was investigated using [^68^Ga] Ga-NOTA-NT-20.3 (**A**), [^64^Cu] Cu-NOTA-NT-20.3 (**B**), and [^55^Co] Co-NOTA-NT-20.3 (**C**).

**Figure 3 pharmaceutics-14-02724-f003:**
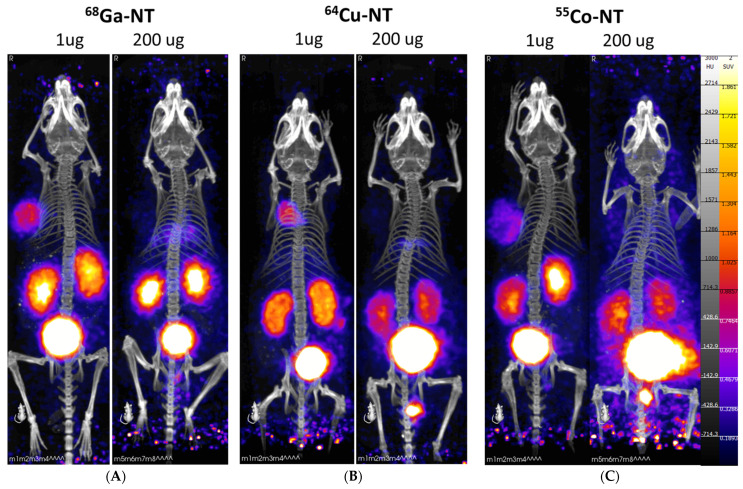
PET/CT imaging 1 h post-injection with radiolabeled NOTA-NT-20.3 in mice bearing HT29 tumor xenografts. Mice were imaged with [^68^Ga] Ga-NOTA-NT-20.3 (**A**), [^64^Cu] Cu-NOTA-NT-20.3 (**B**), and [^55^Co] Co-NOTA-NT-20.3 (**C**).

**Figure 4 pharmaceutics-14-02724-f004:**
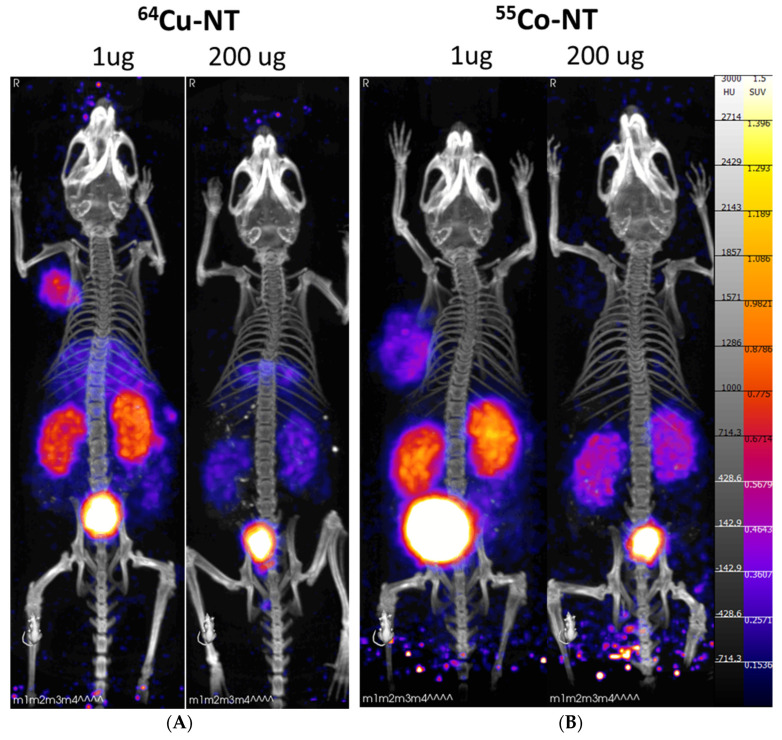
PET/CT imaging 4 h post-injection with radiolabeled NOTA-NT-20.3 in mice bearing HT29 tumor xenografts. Mice were imaged with [^64^Cu] Cu-NOTA-NT-20.3 (**A**) and [^55^Co] Co-NOTA-NT-20.3 (**B**).

**Figure 5 pharmaceutics-14-02724-f005:**
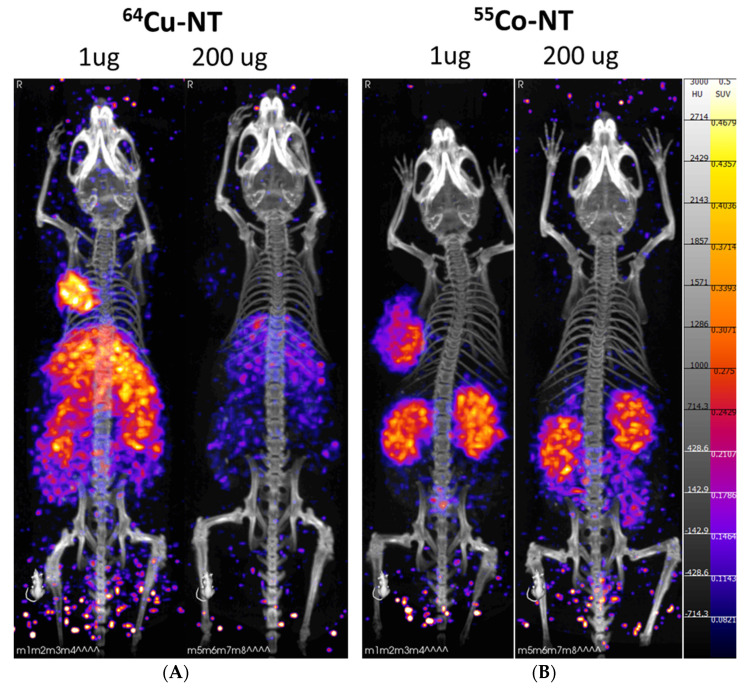
PET/CT imaging 24 h post-injection with radiolabeled NOTA-NT-20.3 in mice bearing HT29 tumor xenografts. Mice were imaged with [^64^Cu] Cu-NOTA-NT-20.3 (**A**) and [^55^Co] Co-NOTA-NT-20.3 (**B**).

**Figure 6 pharmaceutics-14-02724-f006:**
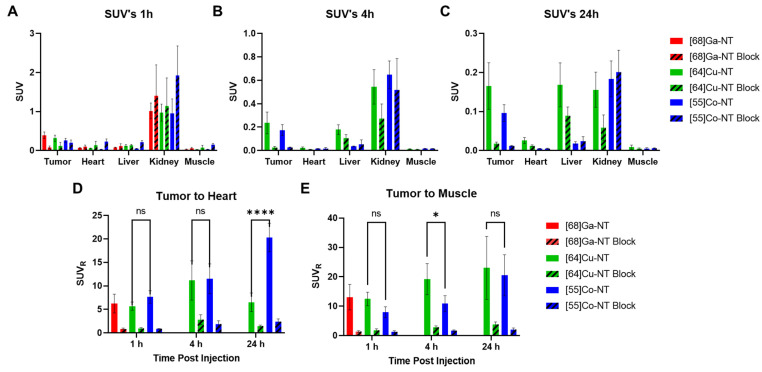
SUV data extracted from the PET/CT images. SUVs were extracted for selected tissues at 1 h (**A**), 4 h (**B**), and 24 h (**C**) post-injection. SUV ratio information was calculated to present tumor to heart ratios (**D**) and tumor to muscle ratios (**E**). ns: not significant, * *p* < 0.05, **** *p* < 0.0001.

**Figure 7 pharmaceutics-14-02724-f007:**
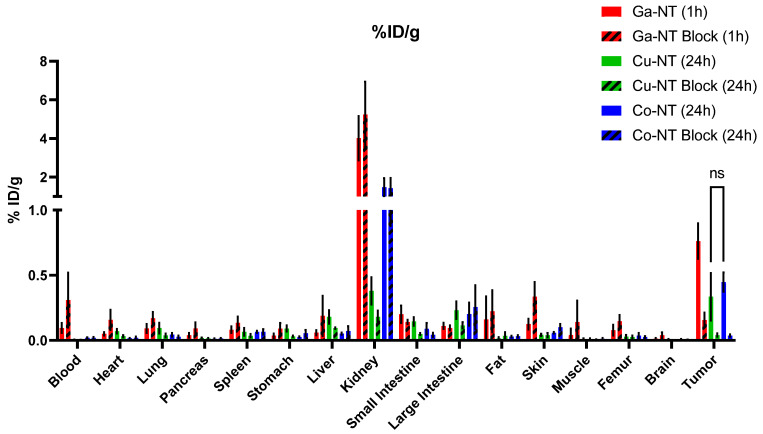
Biodistribution data were calculated after the last imaging timepoint. Organs for the [^68^Ga] Ga-NOTA-NTS-20.3 biodistribution were collected after the 1 h imaging timepoint, whereas for [^64^Cu] Cu-NOTA-NTS-20.3 and [^55^Co] Co-NOTA-NTS-20.3, tissues were collected after the 24 h imaging timepoint.

**Table 1 pharmaceutics-14-02724-t001:** Radionuclides used in this study.

Isotope	Half-Life	Positron Branching Ratio	Mean Positron Energy	Principal Photons (keV)
Ga-68	67.7 min	89%	829 keV	511 (178%), 1077 (3%)
Cu-64	12.7 h	17.5%	278 keV	511 (35.2 %), 1346 (0.5%)
Co-55	17.5 h	76%	570 keV	511 (152%), 931 (75%)

**Table 2 pharmaceutics-14-02724-t002:** Stability of radiolabeled NOTA-NT-20.3 in PBS and human serum.

Isotope	1 h PBS	1 h Serum	4 h PBS	4 h Serum	24 h PBS	24 h Serum
[^68^Ga]Ga-NOTA-NT-20.3	>95%	>95%	N/A	N/A	N/A	N/A
[^64^Cu]Cu-NOTA-NT-20.3	>95%	>95%	>95%	>95%	>95%	>95%
[^55^Co]Co-NOTA-NT-20.3	N/A	N/A	>95%	>95%	>95%	>95%

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
