# Peer review of "PET Imaging of the Neurotensin Targeting Peptide NOTA-NT-20.3 Using Cobalt-55, Copper-64 and Gallium-68"

_pharmaceutics, 2022, doi:10.3390/pharmaceutics14122724_

Round 1

Reviewer 1 Report

The manuscript entitled "PET Imaging of the Neurotensin Targeting Peptide NOTA-NT-20.3 Using Cobalt-55, Copper-64 and Gallium-68" reports a study about NOTA-NT-20.3  being radiolabeled with gallium-68, copper-64, and cobalt-55  for PET imaging. The data is convincible and I enjoy a lot in reading this paper. But some minor revisions should be added to this manuscript to make this paper more publishable.

Comments:

The introduction of this manuscript is too short. The application of isotopes for precise PET imaging in vivo and the off-target effect of isotopes in vivo should be elaborated. Adequate references should be added accordingly.

Author Response

We appreciate the author taking the time to review our manuscript. We have added several paragraphs to the introduction expanding on neurotensin imaging, theranostics, and the impact of radiometal choice upon imaging.

Reviewer 2 Report

This is a well written manuscript that describes the radiolabeling and evaluation of a neurotensin peptide analogue (NOTA-NT-20.3) with various radiometals (Ga-68, Cu-64, and Co-55). The peptides are evaluated in vitro using cell uptake assays as well as subcellular fractionation. In vivo PET imaging is performed with appropriate SUV analysis and biodistribution. In general, this is a sound manuscript with appropriate methods and results that are adequately described. One major issue that should be addressed is the conclusion that the Co-55 construct is superior to the Ga-68 and Cu-64. To this reviewer, the Cu-64 tumor uptake in the images appears to be superior to Co-55 while kidney uptake is greater for Co-55. There are definitely issue with Cu-64 liver uptake compared to Co-55.  This is also seen in the SUV analysis and biodistribution. The only advantage of Co-55 is the tumor to heart SUV at 24 h and maybe tumor to liver ratio?  Other items that should be addressed include:

In the abstract, the authors discuss the peptide being "modified to improve stability". This reads as if this peptide was modified in the paper. It is clear in the introduction that it is a previously studied peptide, but not here. The peptide was first studied approximately 12 years ago so this is not a novel aspect of these investigations. Along these lines, in the results it is stated that the NT-20.3 has "enhanced serum stability compared to endogenous". The stability (half-life) of the endogenous peptide should be stated.

What is the affinity of the peptide for the receptor? Does it change with different radiometals?

The methods about a high specific activity method for Co-55 is confusing. Why show this if it is not in the rest of the manuscript? It seems like Figure 1 should show the HPLC of the Co-55 used for in vitro and in vivo studies. Also, show Ga-68 and Cu-64.

What is the purpose of the subcellular fractionation assays? Is this information useful in radiopharmaceutical design? Is the membrane fraction for the plasma membrane or is it total cellular membranes that would include intracellular membranes?

How many mice were used in each group? I did not see this information.

Statistical section should be added and used to discuss statistical differences for in vitro and in vivo studies.

The blocking of the kidney for the Cu-64 construct is confusing. The authors address this, but it is really not clear what is happening.

Reference 18 does not describe Cu-64 production. 
